# Review of In Situ Hybridization (ISH) Stain Images Using Computational Techniques

**DOI:** 10.3390/diagnostics14182089

**Published:** 2024-09-21

**Authors:** Zaka Ur Rehman, Mohammad Faizal Ahmad Fauzi, Wan Siti Halimatul Munirah Wan Ahmad, Fazly Salleh Abas, Phaik Leng Cheah, Seow Fan Chiew, Lai-Meng Looi

**Affiliations:** 1Faculty of Engineering, Multimedia University, Cyberjaya 63100, Malaysia; 1211400112@student.mmu.edu.my (Z.U.R.);; 2Institute for Research, Development and Innovation (IRDI), IMU University, Bukit Jalil, Kuala Lumpur 57000, Malaysia; 3Faculty of Engineering and Technology, Multimedia University, Bukit Beruang, Melaka 75450, Malaysia; fazly.salleh.abas@mmu.edu.my; 4Department of Pathology, University Malaya-Medical Center, Kuala Lumpur 50603, Malaysia; cheahpl@ummc.edu.my (P.L.C.); looilm@ummc.edu.my (L.-M.L.)

**Keywords:** deep learning, pathologies, human epidermal growth factor receptor 2 (HER2), fluorescent in situ hybridization (FISH), silver-enhanced in situ hybridization (SISH)

## Abstract

Recent advancements in medical imaging have greatly enhanced the application of computational techniques in digital pathology, particularly for the classification of breast cancer using in situ hybridization (ISH) imaging. HER2 amplification, a key prognostic marker in 20–25% of breast cancers, can be assessed through alterations in gene copy number or protein expression. However, challenges persist due to the heterogeneity of nuclear regions and complexities in cancer biomarker detection. This review examines semi-automated and fully automated computational methods for analyzing ISH images with a focus on *HER2* gene amplification. Literature from 1997 to 2023 is analyzed, emphasizing silver-enhanced in situ hybridization (SISH) and its integration with image processing and machine learning techniques. Both conventional machine learning approaches and recent advances in deep learning are compared. The review reveals that automated ISH analysis in combination with bright-field microscopy provides a cost-effective and scalable solution for routine pathology. The integration of deep learning techniques shows promise in improving accuracy over conventional methods, although there are limitations related to data variability and computational demands. Automated ISH analysis can reduce manual labor and increase diagnostic accuracy. Future research should focus on refining these computational methods, particularly in handling the complex nature of HER2 status evaluation, and integrate best practices to further enhance clinical adoption of these techniques.

## 1. Introduction

Breast cancer remains the most prevalent malignancy among women worldwide, with over 2 million new cases and nearly 630,000 deaths reported in 2018 alone [1]. The high morbidity and mortality rates associated with breast cancer have propelled research aimed at improving histopathologic image-based computational techniques. These techniques have become essential for identifying cancer subtypes, which are critical for clinical decision making and personalized treatment strategies. Digital pathology, powered by advancements in imaging and computational capabilities, has emerged as a promising field to support the precise and efficient classification of breast cancer.

The development of whole-slide digital imaging, combined with the growing importance of tissue-based biomarkers for therapy stratification, has greatly expanded the applications of image analysis in digital pathology. Notably, techniques such as hematoxylin and eosin staining (H&E), immunohistochemistry (IHC), and in situ hybridization (ISH) are regularly employed for visualizing and analyzing tissue samples. H&E remains the cornerstone of histopathology, providing detailed cellular and tissue architecture, while IHC targets specific proteins, aiding in the functional interrogation of tissues. ISH, which focuses on the detection of nucleic acid sequences, offers deeper insights into gene expression, especially in cases where protein expression is insufficient or ambiguous [2].

In breast cancer diagnostics, ISH is particularly valuable for detecting *HER2* gene amplification, a critical prognostic and predictive marker for about 20–25% of breast cancers. This amplification is commonly assessed through HER2 and CEP17 (centromere enumeration probe for chromosome 17) signals, providing a quantitative measure for determining HER2 status [3]. Although fluorescence in situ hybridization (FISH) is considered the gold standard for HER2 testing, alternative methods such as chromogenic ISH (CISH) and silver-enhanced ISH (SISH) have been developed to offer cost-effective solutions compatible with bright-field microscopy (Figure 1 and Table 1).

Despite these advancements, the interpretation of ISH images remains a complex and time-consuming task, often requiring manual analysis by experienced pathologists. The emergence of computational techniques—ranging from traditional image processing algorithms to deep learning models—has the potential to automate this process, improving diagnostic accuracy, efficiency, and reproducibility [6]. The availability of large, digitized histopathology datasets has accelerated the application of these computational methods, yet challenges such as data annotation and variability in staining techniques persist.

This review aims to provide a comprehensive overview of semi-automated and fully automated ISH-based computational methods, with a particular focus on breast cancer classification. We review the literature from 1997 to 2023, with an emphasis on image processing techniques and machine learning models, particularly deep learning, as they apply to *HER2* gene amplification detection. Figure 2 provides a high-level breakdown of the computational methods commonly applied in the field. Through this review, we aim to clarify the relationships between different computational approaches, highlight key advancements, and discuss the potential for integrating these methods into routine pathology workflows.

### 1.1. Inclusion and Exclusion Criteria

To ensure the relevance and specificity of this review, the following criteria were used to include or exclude studies:**Inclusion Criteria:** Studies that applied artificial intelligence (AI), machine learning (ML), or deep learning (DL) techniques to the analysis of ISH images, specifically for *HER2* gene amplification detection.**Exclusion Criteria:** Papers that focused on other pathology stains (e.g., H&E, IHC) or did not involve computational techniques.

This rest of this paper is structured as follows: Section 1.3 explores key challenges in ISH image analysis. Section 1.6 reviews the state-of-the-art methodologies in computational ISH analysis. Section 2 outlines common computational techniques for image processing in pathology. Finally, Section 5 summarizes the findings and outlines recommendations for future research directions.

### 1.2. In Situ Hybridization (ISH)

In situ hybridization (ISH) is a cytogenetic technique that allows for the detection, quantification, and localization of specific nucleic acid sequences within cells or tissues at high resolution. This method plays a pivotal role in understanding the organization, regulation, and function of genes by revealing the physical positions of DNA or RNA sequences on chromosomes or within tissues. ISH works by hybridizing a labeled probe—complementary to the target nucleic acid—with the DNA or RNA of the tissue or chromosome under examination. The types of probes used for DNA and RNA have been comprehensively described in earlier studies [7]. Probes can be labeled chemically or radioactively, and this labeling allows for the precise detection of hybridization signals.

In the context of *HER2* gene amplification, ISH techniques such as FISH, CISH, and SISH are routinely used to determine gene copy number alterations, which are critical for evaluating HER2 status in breast cancer (as shown in Table 1). Each method offers varying advantages in terms of sensitivity, ease of use, and cost-effectiveness, with SISH being particularly suitable for bright-field microscopy applications.

### 1.3. Challenges of In Situ Hybridization

The ISH process presents numerous challenges, both biological and technical, that complicate the analysis of the resulting images. These challenges impact the accuracy, reproducibility, and scalability of computational approaches used for automated analysis.

#### 1.3.1. Technical Challenges

**Signal variability**: ISH images often exhibit significant variability in signal intensities, not only between the target and non-target cells but also within different regions of the same tissue sample. This inconsistency complicates accurate signal detection and segmentation [8].**Complex tissue structures**: ISH images often include a mixture of cell populations and complex tissue architectures, making it difficult to isolate and analyze regions of interest. Overlapping or closely spaced signals, particularly in multi-probe ISH experiments, further add to this complexity.**Large image size**: Whole-slide ISH images can be very large, requiring significant computational power for storage, processing, and analysis. Multi-channel ISH images with multiple probes introduce additional layers of complexity to the segmentation and classification tasks [9].**Tissue preparation**: The requirement for very thin tissue sections (typically 3–7 μm in thickness) introduces potential artifacts to the images, such as tearing or folding, which can distort the analysis [9].

#### 1.3.2. Biological Challenges

**Heterogeneous tissue samples**: The biological complexity of tissues introduces variability in cell types, gene expression patterns, and tissue structures [10]. This heterogeneity can lead to uneven distribution of hybridization signals, further complicating segmentation and quantification tasks.**Overlapping signals**: In biological samples, signals from adjacent cells or closely located genes often overlap, making it difficult to accurately assign signals to specific cells or chromosomes [11].**Non-specific staining**: Background noise and non-specific staining are common in ISH images, reducing the contrast between the target signal and the background. This interferes with the ability of automated systems to distinguish true signals from artifacts, especially in low-signal regions [11].

### 1.4. Data Acquisition

Accurate and reproducible ISH analysis requires well-optimized protocols for data acquisition, starting from tissue preparation to probe hybridization. In our study, the INFORM HER2 DNA and CEN17 probes were replaced with the Ventana HER2 siler ISH Probe Cocktail, applied using the Ventana Benchmark automated device [12]. This method streamlines the process, reducing manual intervention and improving consistency across samples.

The data acquisition process for SISH can be outlined as follows:1.**Sample preparation**: Tissue samples are baked at 60 °C for 20 min to ensure proper adhesion to the slides.2.**Probe hybridization**: The HER2 DNA and chromosome 17 probes are denatured at different temperatures and hybridized with the target sequences.3.**Stringency washes**: Stringent washing is performed to remove any unbound probes, ensuring high specificity of the hybridization signals.4.**Signal detection**: The ultraView SISH Detection Kit is used for visualizing the HER2 and CEP17 probes, with silver deposition providing contrast for bright-field microscopy analysis.5.**Counterstaining**: Hematoxylin is applied as a counterstain to enhance visualization under a light microscope.

Compared to traditional FISH methods, the use of SISH significantly reduces the overall time required for analysis (from 12–16 h to 6 h) and can be performed using a standard light microscope, making it more accessible for routine pathology laboratories.

### 1.5. Probe Design and Labeling Techniques

The sensitivity and specificity of ISH rely heavily on the design of the probe and its labeling technique. Probes can be classified based on their labeling method:**Radiolabeled probes**: These probes use radioactive isotopes to tag the nucleic acid sequence of interest, offering high sensitivity but requiring specialized equipment for detection and posing health risks [13].**Non-radioactive probes**: Modern techniques such as biotin or digoxigenin labeling have become more popular, offering safer alternatives that use colorimetric or fluorescent detection methods [14].**Direct enzyme labeling**: Enzyme-conjugated probes catalyze colorimetric reactions, offering a straightforward way to visualize hybridization signals without the need for secondary detection steps [15].

The development and selection of appropriate probes are critical for ensuring accurate *HER2* gene amplification analysis. Ongoing research is focused on improving the sensitivity of these probes to detect smaller genetic aberrations, expanding the potential clinical applications of ISH techniques.

### 1.6. From Glass Slide to Whole-Slide Image

The transformation of traditional glass slides into whole-slide images (WSIs) has revolutionized the field of digital pathology, providing pathologists with the ability to view, analyze, and share high-resolution tissue images. However, this transition requires the precise control of the entire image acquisition pipeline, from tissue processing and staining to scanning and image quality assurance. Table 2 summarizes some of the key challenges in standardizing this process for clinical and research applications.

The digitization of histological slides enables advanced computational analysis, including segmentation, feature extraction, and classification tasks that form the backbone of AI-based ISH image analysis.

### 1.7. HER2 Status Evaluation

The amplification or overexpression of the HER2 oncogene is observed in approximately 20% of invasive breast carcinomas [16]. This amplification is associated with poor prognosis, necessitating targeted therapies such as trastuzumab [17]. The accurate assessment of the HER2 status is critical for making personalized therapeutic decisions in breast cancer patients [18]. When IHC results are equivocal, such as a 2+ expression level [19], further analysis using ISH is performed to confirm HER2 amplification. Pathologists commonly follow ASCO/CAP guidelines, using methods such as FISH, CISH, and SISH to compare HER2 signals with CEP17 (chromosome 17 centromere) signals [20].

In the DISH procedure, pathologists manually count HER2 (black) and CEP17 (red) signals under a microscope. A total of 20 cells are typically counted, and if the HER2/CEP17 ratio is borderline, an additional 20 cells are counted for more accurate results [21]. However, challenges arise in interpreting borderline and heterogeneous tumors, where HER2-amplified cells may be concentrated in specific areas or mixed with non-amplified cells [22,23]. The presence of CEP17 polysomy further complicates interpretation, as it can inflate the HER2/CEP17 ratio without indicating true amplification [24]. Subjectivity in cell selection and technical variability across laboratories also impact the consistency of the results [25,26].

### 1.8. Current Evaluation Practice

Manual HER2 ISH evaluation is a time-consuming and subjective process, heavily reliant on the pathologists’ selection of representative cells, which may introduce selection bias [27,28]. Various clinical guidelines, including cutoffs for signal counts, ratios, and the fraction of amplified cells, contribute to the complexity of assessing equivocal or heterogeneous cases. Despite these challenges, the manual evaluation process remains widespread, though it is not yet adequately standardized. Investigations into automated ISH evaluation have revealed discrepancies in the sampling methods, from small TMA cores to full tissue sections [29], with sample sizes ranging from a few fields of view to 20–60 nuclei per case [30,31].

### 1.9. Toward Computational Digital Pathology

Efforts to automate HER2 ISH testing using computational methods have gained attraction in recent years [29,32], with digital image analysis offering the potential to reduce the pathologist’s workload and enhance diagnostic precision. While good-quality samples and standardized procedures are still necessary, image analysis can serve as a valuable decision-support tool [29,30]. The computer-assisted quantification of FISH signals has shown promise, particularly in improving the evaluation of equivocal and heterogeneous cases through large-scale sampling and unbiased analysis [29,31].

Despite the advancements in digital pathology for FISH and CISH stains, limited research has been conducted on SISH stains for HER2 scoring and amplification using computational methods. Our research aimed to explore the potential of high-resolution digital HER2 SISH images to generate objective, statistically derived indicators of intratumoral heterogeneity in the HER2 status. Such efforts are crucial for improving the accuracy and scalability of HER2 amplification evaluation in clinical settings.

## 2. Computational Digital Pathology

While many healthcare and life sciences organizations recognize the potential of using artificial intelligence to analyze whole-slide images (WSIs), developing an automated slide analysis pipeline presents significant challenges. A functional WSI pipeline must handle a high volume of digitized slides at low cost and with high efficiency. Computational image analysis generally involves several key steps, which are discussed in this section. Figure 3 illustrates our proposed scheme for these steps.

Digital pathology has become central to both research and clinical diagnostics, driven by advancements in imaging technology and the availability of efficient computational tools. WSIs have been instrumental in this transformation, allowing for the rapid digitization of pathology slides into high-resolution images.

### 2.1. Data Preprocessing

Image preprocessing typically involves the following steps:**Noise reduction and artifact elimination**: Removing irrelevant or non-informative data, such as slide backgrounds, dust particles, or scanning artifacts.**Dataset consistency**: Ensuring the creation of a standardized and consistent dataset by eliminating variations across different samples.**Tiling for deep learning models**: Most deep learning models cannot process gigapixel images directly. Therefore, WSIs are split into smaller tiles, which are processed in batches during downstream modeling.

Preprocessing is critical for using computational resources efficiently and minimizing errors caused by noise or artifacts in the images. Tissue segmentation algorithms often rely on effective preprocessing, as irrelevant variations can disrupt accurate image analysis. Morphological transformations, frequently used in image postprocessing, are also employed during preprocessing to detect and remove artifacts.

Automated image analysis in digital pathology depends on the visual quantification of image features. Pathologists use tissue segmentation algorithms based on this initial preprocessing step [33]. Signal estimation, or saturation, is a common optical effect that occurs when scanner software exceeds its recognition threshold for certain pixel values—such as when detecting overexpressed genes. Yang et al. [34] proposed a mixture-based model for spot segmentation that addresses this issue by estimating dense pixel values with a censored component. Table 3 provides an overview of commonly used preprocessing techniques, their applications, and constraints.

Data preprocessing plays a critical role in ensuring the success of downstream modeling for whole-slide image analysis. Preprocessing not only reduces noise but also prepares the images for feature extraction, segmentation, and classification tasks. Understanding the properties and limitations of each technique is essential for developing robust, scalable pipelines in computational digital pathology.

### 2.2. Feature Extraction

Table 4 lists various feature extraction methods used in image analysis. Histopathology images often rely on pathologists’ clinical experiences to guide feature extraction techniques. As a result, property-based features are used as a foundation. This section covers three main types of features: shape-based, texture-based, and color-based. Each feature type is detailed in the following subsections, and Table 4 provides brief descriptions of the feature-wise performances for some methods.

#### 2.2.1. Shape-Based Features

The classification of pathology images often relies on the morphology of nuclei and cell sections. Shape and size (morphology) play a crucial role in diagnosing lesions and cancers. Spherical or quasi-spherical shapes are easier to characterize as feature vectors than more complex, naturally occurring cell shapes. Shape features can quantify the cell or nucleus region by calculating attributes such as size, area, and perimeter.

For example, Ref. [56] represented color distribution in 3D cervical cancer images using intensity data and shape details. A large annotated dataset of histological images related to the cervix, vagina, and uterus was used in [57] to assess the quality of shape features, such as rotation-invariant features. Shape-based features were also applied in [58] for cervical cancer detection using unsupervised k-means clustering and geometric feature extraction from spanning tree graphs.

Shape features have also been applied in cancer cell detection using FISH spots [39], where automated detection and classification rules were employed to identify and count FISH spots accurately. Similarly, Ref. [42] extracted lymphocytes from plasma and used shape and contour features for leukemia diagnosis, while [43] introduced contour signature and fractal features for classifying lymphocyte nuclei in leukemia cases.

Various shape-based extraction methods used for image analysis are described in Table 4.

#### 2.2.2. Texture-Based Features

Texture is a crucial feature for analyzing spatial patterns and tissue organization in pathology images. Texture analysis has been widely employed for classification tasks, particularly in pathological image analysis [59]. Texture patterns can range from pixel-level patterns to larger structures that capture spatial relationships.

For example, ref. [60] proposed using fractal texture features based on optical density surface areas for analyzing cervical cell images. Texture features were also effective for detecting developmental phases in ISH images of gene expression patterns in [41], where texture factors provided insights into Drosophila gene patterns.

Additionally, local binary patterns (LBPs) were used in [10] to analyze ISH images and train gene classifiers for different layers of the cerebellum. Texture features have also been applied for HER2 2+ status evaluation, where 279 texture features were extracted from FISH images [52], and hyperspectral image compression techniques were explored in [47] for texture-based segmentation and classification.

Table 4 lists several texture-based feature extraction methods applied in image analysis.

#### 2.2.3. Color-Based Features

Color is one of the most widely used features in digital pathology for selecting or rejecting cell sections. Color features are extracted in various color spaces, including RGB, to analyze images. However, color representation varies across devices, and standardization is essential for accurate analysis [61]. In pathology, color features help distinguish cells, tissues, and other structures.

For example, Ref. [43] used color segmentation to analyze blood cells in leukemia diagnosis. Blood and bone marrow smears from patients with acute lymphoblastic leukemia were analyzed in [48] using a k-means clustering approach, and the resulting color features were used for classification.

In M-FISH image analysis, color features are employed for chromosome classification [45], and SVM classifiers have been used for distinguishing leukemic white blood cells based on color features [53]. Various color-based extraction methods are summarized in Table 4.

### 2.3. Segmentation

Segmentation is a critical task in image analysis, used to isolate regions of interest (ROIs) such as cell nuclei, tissues, or tumor areas. Segmentation techniques can vary from threshold-based methods to more advanced approaches like region-based or machine learning techniques. Effective segmentation is vital for accurate feature extraction and classification.

Table 5 summarizes various segmentation techniques, categorized by application (e.g., nuclei segmentation, cancer cell detection, tumor area detection).

#### 2.3.1. Thresholding-Based Segmentation

Thresholding is one of the most common segmentation techniques, particularly for grayscale and RGB images. In threshold-based segmentation, pixel intensity is used to create image sections, which are then analyzed based on intensity differentials. Adaptive thresholding methods, such as Otsu’s method [69], are widely used to enhance segmentation accuracy.

For example, Ref. [70] proposed an intelligent framework for FISH data analysis using a hybrid nuclei segmentation technique. Threshold-based segmentation methods have also been applied for nuclei segmentation in HER2 status detection [62], where contrast enhancement and thresholding were used for improving image quality.

Threshold-based segmentation techniques for ISH images, including examples of CISH, FISH, and SISH images, are illustrated in Figure 4.

#### 2.3.2. Region-Based Segmentation

In region-based segmentation, pixels are grouped based on intensity and spatial connectivity. This method works well for images with distinct regions, but multisegment images may require more processing power. Clustering methods like fuzzy c-means are often used for the soft clustering of pixels into multiple regions [71].

In [72], clustering-based segmentation was applied for identifying tumor regions, and machine learning techniques have also been integrated for region-based segmentation tasks [73,74]. Examples of region-based segmentation techniques applied to nuclei and tumor detection are listed in Table 5.

### 2.4. Classification

This section discusses the methods used for classifying ISH pathology images. These methods are categorized into two main subcategories: conventional classification, discussed in Section 2.4.1, and deep learning methods, covered in Section 2.4.2.

#### 2.4.1. Classification through Conventional Methods

The first developments in computer vision date back to the 1960s, and the field has since become an essential part of intelligent systems in industries such as security, robotics, autonomous vehicles, and medical imaging [75]. In digital pathology, the task of classifying pathology images involves assigning biomarkers to different classes based on image input. Conventional computer vision methods leverage features such as color, shape, texture, and size to perform classification, making use of RGB images to detect disease-specific patterns [76].

Table 6 provides a comparison of conventional and deep learning methods for ISH image classification, highlighting their respective pros and cons.

Machine learning models such as decision trees, neural networks (NNs), K-nearest neighbors (KNN), and support vector machines (SVMs) have been widely applied in pathology classification tasks [77,78,79]. Each method offers distinct advantages: SVMs handle linear and non-linear data mapping using kernel functions, decision trees provide a probability-based graph for multi-class classification, and KNN is a non-parametric method that learns from data indefinitely.

For example, Hongbao et al. [45] used minimal representation-based classifiers to enhance chromosome analysis for cancer and genetic disease diagnostics using M-FISH images. Improved segmentation techniques and ensemble classifiers have also been applied in the diagnosis of acute lymphoblastic leukemia (ALL) [46].
diagnostics-14-02089-t006_Table 6Table 6A comparison of conventional and deep learning methods used in digital pathology for ISH image classification.YearISH StainML/DLPros and ConsRef.2012M-FISH✓ (ML )**Pros:** Effective for small datasets, interpretable models. **Cons:** Limited scalability and feature extraction capability. [45]2014Leukemia✓ (ML)**Pros:** Simple, computationally efficient for screening. **Cons:** Handcrafted features may miss complex patterns. [46]2016FISH✓ (DL)**Pros:** Automated feature extraction, scalable. **Cons:** Requires large datasets and computational power. [80]2017ISH✓ (DL)**Pros:** Learns hierarchical features from raw images. **Cons:** Black-box models, high computational requirements. [81]2018Monoclonal antibody WSIs✓ (DL)**Pros:** High accuracy, effective for complex features like cell membranes. **Cons:** Training requires large amounts of annotated data. [82]2019ISH✓ (DL)**Pros:** Learns from raw pixel data. **Cons:** Struggles to interpret feature representations. [83]2020CISH✓ (ML)**Pros:** Cost-effective, interpretable. **Cons:** Lower accuracy than DL methods for complex data. [84]2021ISH✓ (DL)**Pros:** Can handle large image datasets; **Cons:** Black-box model, interpretability challenges. [85]Note: ML stands for machine learning, and DL for deep learning. The table highlights the advantages (pros) and limitations (cons) of both approaches in terms of scalability, interpretability, and computational cost.


Liew et al. [80] applied classification-based methods for FISH image analysis, while [84] explored CISH image classification using Haralick texture features and principal component analysis (PCA) for dimensionality reduction. Table 6 summarizes recent studies on ISH image-based pathology disease classification.

#### 2.4.2. Classification through Deep Learning

Deep learning has revolutionized image analysis in recent years, particularly with the use of convolutional neural networks (CNNs) for medical imaging tasks [86,87]. CNNs apply convolutional filters to input images, learning hierarchical feature representations automatically, without the need for manual feature extraction. This makes CNNs especially powerful for tasks such as pathology image classification.

CNNs are trained on large datasets, allowing them to learn from raw pixel data and optimize for high-level semantic features such as cell boundaries and biomarker signals [88,89]. For example, Ref. [81] proposed a deep convolutional denoising autoencoder (CDAE) for constructing compact ISH image representations, while [82] introduced Her2Net, a deep learning framework for HER2-stained breast cancer image analysis, which includes cell membrane and nucleus detection, segmentation, and classification.

Similarly, Ref. [83] employed autoencoders and convolutional neural networks for learning feature representations directly from image pixels, demonstrating the superiority of these methods over traditional feature extraction techniques. Transfer learning strategies were also explored to adapt pretrained models to ISH images, improving accuracy in biomarker detection and disease classification.

The reference work for ISH image-based pathology disease classification is summarized in Table 6.

## 3. Image Analysis on SISH

In the realm of HER2 determination, SISH has emerged as a viable alternative to traditional methods like FISH and CISH [90]. SISH [91,92] represents a novel approach that leverages bright-field imaging, similar to CISH, and has been significantly enhanced by advancements in automation. The Ventana Medical System (Tucson, AZ, USA) has developed a fully automated system that improves the efficiency and consistency of bright-field in situ hybridization, thereby reducing the risk of human error. This system allows for the automated detection of chromogenic signals, enabling the simultaneous running of HER2 and CEP17 assays on related tissue slides.

In line with the ASCO/CAP guidelines, the evaluation of *HER2* gene amplification status using SISH was conducted in a blinded fashion. The analysis involved examining 20 non-overlapping nuclei for HER2/CEP 17 signals and calculating the HER2/CEP 17 ratio. A ratio greater than 2.2 indicates *HER2* gene amplification, while a ratio of 1.8 or less suggests a lack of amplification. Ratios between 1.8 and 2.2 are deemed equivocal, necessitating the counting of signals from an additional 20 tumor nuclei in a second target area to compute a new ratio. Benign breast epithelial cells and other adjacent benign cells served as internal controls throughout the process.

This study’s focus on SISH was not only to validate its efficacy in HER2 status assessment but also to lay the groundwork for more advanced computational analysis. Moreover, the combination of SISH with automated image analysis presents an opportunity to create scalable solutions capable of analyzing high-throughput HER2 assays, minimizing the variability often observed in manual counting techniques. As discussed in our review paper, SISH offers a promising platform for integrating computational techniques to enhance the accuracy and scalability of HER2 analysis. The adoption of SISH in computational pathology is particularly significant, given its compatibility with automated image analysis systems, which are essential for handling the increasing volume and complexity of digital pathology data.

**Software Information:** Image analysis was performed using MATLAB R2021b (MathWorks, Natick, MA, USA) was used to create custom algorithms. Both software tools were downloaded from their official websites, with MATLAB accessed from https://www.mathworks.com/products/matlab.html, accessed on 29 July 2024.

## 4. Limitations of the Research Work

This study faced several limitations, including variability in tissue samples and heterogeneity in ISH images, which may lead to inconsistencies in image analysis. The reliance on SISH, a newer and less widely adopted staining technique, could limit the generalizability of the results. Additionally, the computational demands required for processing large whole-slide images (WSIs) pose scalability challenges. Future work could focus on improving algorithms for tissue heterogeneity and utilizing federated learning across clinical institutions to address the variability in datasets. Further refinement is needed to address complex cases of HER2 heterogeneity and to validate the findings across diverse datasets and clinical settings.

## 5. Conclusions

In this paper, we have provided a comprehensive overview of the advancements in machine learning and in situ hybridization (ISH) image analysis using computational methods. We began with an introduction to ISH images and the associated challenges, followed by a discussion of ISH-related work and the application of computational methods in ISH stain pathology image analysis. The computational image analysis section encompasses image data acquisition, preprocessing, segmentation, feature extraction, and classification. We assessed relevant works based on their specific technical categories under each application goal from a computational pathology perspective. By reviewing all related studies on ISH stains using computational image analysis methods, we identified the most popular image feature extraction, segmentation techniques, and classification approaches.

Machine vision techniques in this sector have demonstrated a consistent overall development trend, albeit with a cautious approach. The most cutting-edge technologies in this field typically emerge three to five years later compared to other domains. This “slow starter” phenomenon is primarily due to the interdisciplinary nature of the research, where machine vision scientists often have limited knowledge of ISH stain pathology. However, as more biomedical engineering students are educated, we expect the progress of machine vision techniques in this field to synchronize with those in other domains.

Machine vision techniques have evolved significantly in ISH image analysis, though the interdisciplinary nature of pathology, and AI often leads to a slower adoption of these technologies. As these fields converge, we anticipate that digital pathology, with integrated computational methods, will enhance the accuracy and reproducibility of HER2 assessments, ultimately improving patient outcomes through personalized cancer treatment.

Furthermore, the machine vision approaches discussed in this paper can be applied to various microscopic image analysis disciplines beyond ISH digital pathology. Recent rapid advancements in this field have shifted the debate around digital pathology, enabling greater accuracy and efficiency through computational pathology. While the potential of powerful new models to support clinicians in decision making is promising, translating these models into medical practice remains challenging. Digital pathology, distinguished by its comprehensive image acquisition process, often involves subsampling or selecting small tiles from a large whole-slide image (WSI), either systematically or randomly.

Figure 5 illustrates an example of computer-based nuclei and HER2 detection from a SISH pathology image. Detecting breast cancer using the HER2 ratio with SISH stains is complex and poses significant challenges for the automatic localization of tumor regions in SISH WSI images. HER2 scoring follows a specified procedure, which includes several key points and challenges:Selecting appropriate regions with more red and black signals from the SISH WSI stain image.Localizing the nuclei region, which is difficult due to the fusion of nuclei in many areas of the WSI.Choosing 20 nuclei with signals and discarding faint nuclei.After selecting 20 nuclei, separating the red and blue signals, ensuring that two identical signals are not fused.

Due to these challenges, manually identifying HER2 scoring from SISH stains is not easy. Computational techniques are necessary to automatically compute the HER2 score from SISH stains.

### Future Directions

ISH is a unique tissue image-based molecular analysis method used for the precise microscopic detection and localization of DNA, mRNA, and microRNA in metaphase spreads as well as in cell and tissue preparations. In comparison, IHC (immunohistochemistry) is invaluable for the localization, detection, and quantification of antigens, including HER2 signals. Thus, deploying automated machine learning techniques with ISH holds significant promise for the future. Artificial intelligence (AI) provides a powerful tool for extracting information from ISH digitized whole-slide images (WSIs). Numerous techniques have been developed to address diverse tasks related to HER2 scoring from ISH stain images using machine learning methods.

In the future, pathological imaging and machine vision technologies should be developed together organically, with features such as real-time pathological image processing under a microscope or endomicroscopy (e.g., virtual staining and class labeling of pathological images). Emerging AI models, such as transformers and self-supervised learning techniques, offer significant promise in overcoming current challenges in real-time image processing and live diagnosis, moving pathology closer to integrated, fully automated solutions. Microscopes equipped with apps or software for the image analysis of diseased samples can be fitted with small, high-performance CPU processors. Pathologists will be able to monitor cells or tissue types in their actual range of vision and decide if they are normal or abnormal in real time through these systems. They can also witness and observe a number of virtual stained images created using basic lens staining. Simultaneously, the related data analysis report and virtual staining image will be transmitted to a specified mobile phone, computer, or mailbox, enabling real-time scoring. There are numerous effective and novel strategies that can be used to attain these objectives.

## Figures and Tables

**Figure 1 diagnostics-14-02089-f001:**
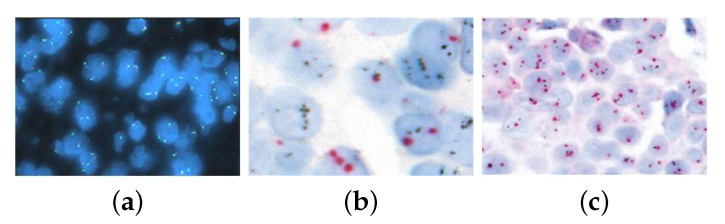
Differenttypes of cytogenic images resulting from ISH: (**a**) FISH at 20× magnification, (**b**) CISH at 20× magnification, and (**c**) SISH at 40× magnification.

**Figure 2 diagnostics-14-02089-f002:**
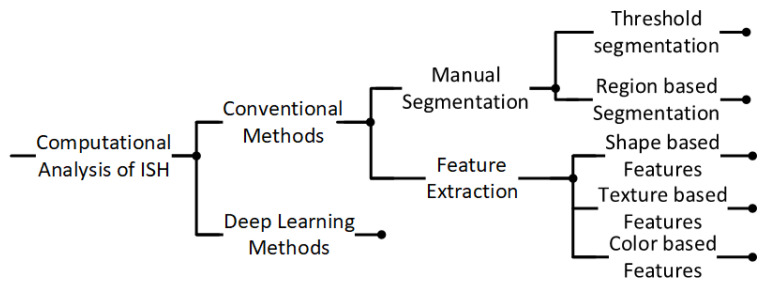
Breakdown of computational methods commonly used for histopathology image analysis.

**Figure 3 diagnostics-14-02089-f003:**
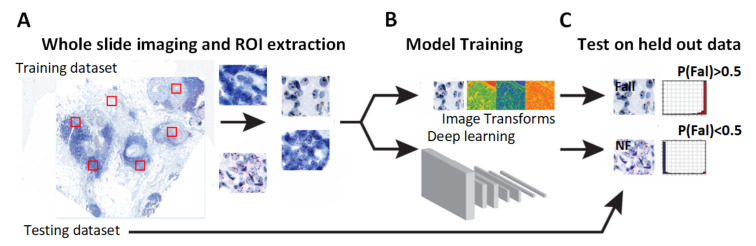
A machine vision-based approach used in digital pathology image analysis. The red squares in subfigure (**A**) indicate selected regions for machine vision analysis. The whole slide image (WSI) is at a magnification level of 40×.

**Figure 4 diagnostics-14-02089-f004:**
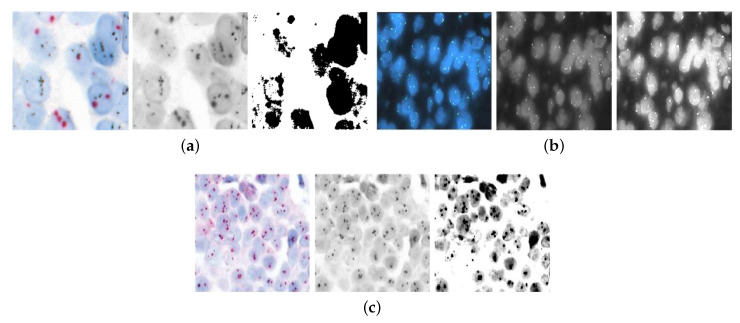
Examples of how digital photographs have been altered using grayscale-based contrast enhancement and thresholding for different cytogenetic types of ISH: (**a**) scale variation in CISH at 20× magnification, (**b**) scale variation in FISH at 20× magnification, and (**c**) scale variation in SISH at 40× magnification.

**Figure 5 diagnostics-14-02089-f005:**
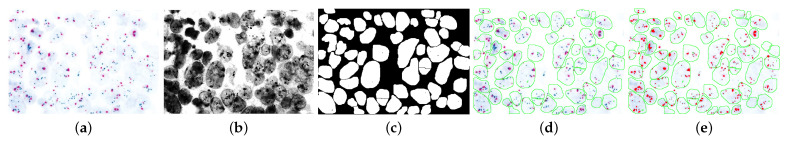
Automated image processing-based system demonstration at 40× magnification: (**a**) original SISH images, (**b**) preprocessed for ground truth generation, (**c**) nuclei-labeled ground truth images, (**d**) marked labeled nuclei on the original image, and (**e**) marked labeled nuclei and HER2 signals. More precise signal detection refines nuclei segmentation.

**Table 1 diagnostics-14-02089-t001:** ISH applications for HER2 amplifications based on different chromogenic systems.

Technique	Target	Explanation	Ref.
FISH	*HER2* gene/CEP17	Fluorescence in situ hybridization (FISH) uses fluorescent probes to detect *HER2* gene amplification and chromosome 17 centromere (CEP17) in tumor cells.	[3]
CISH	*HER2* gene/CEP17	Chromogenic in situ hybridization (CISH) uses chromogenic probes that produce a colorimetric reaction, making it easier to view *HER2* gene amplification and CEP17 under a regular microscope.	[4]
SISH	*HER2* gene	Silver-enhanced in situ hybridization (SISH) is similar to CISH but uses silver deposition to visualize *HER2* gene amplification, allowing the use of standard bright-field microscopy.	[5]

**Table 2 diagnostics-14-02089-t002:** Current computer-based image analysis limitations and potential solutions.

Issue	Problem	Proposed Solution
Tissue analysis and its standardization	Processing of variabilities and tissue harvesting	Revision of histology techniques across centers to improve quantitative analysis downstream. Subspecialty societies are involved
Image analytics	Variability in scanners and image problems	Standard quality assurance and calibration methods are implemented to check the image linearity, uniformity, and reproducibility
Data integration	Extraction of data, spanning multiple length scales, representation, and fusion	Data gathering and storage should be standardized. Development of ontologies. New data fusion methods are being developed

**Table 3 diagnostics-14-02089-t003:** Descriptions of common image preprocessing techniques.

Technique	Description	Applications	Constraints
Elementary processing [33,35]	Signal processing filters are used to process a group of adjacent pixels	Smoothing and gradient analysis for better edge detection	Limited for complex and non-linear signal processing
Intensity estimation [34,36]	The estimation of missing pixel values using spatial and non-spatial analysis	Noisy pixel value determination in grayscale and RGB images	Non-uniform object lighting may require prior knowledge
Geometric estimation [37]	Geometric distortion estimation using relative motion, angle, speed, and 2D to 3D representation	Geometric detail determination in mobile robotics and remote sensing applications	The sensor and object angle, location, and relative speed must be known
Holistic processing [38]	A set of filters are used for convolution for image restoration	Identifying holistic image features	Requires complex stochastic analysis and prior knowledge

**Table 4 diagnostics-14-02089-t004:** Propertiesof feature descriptors.

Year	Ref.	Image & Stain Type	SF	CF	TF	Feature Description	Accuracy
2009	[39]	FISH	✓	✗	✗	Size, circularity, and compactness were computed	96.90%
	[40]	ISH	✗	✓	✗	Anti-digoxigenin (DIG) and fluorescein-labeled riboprobes	–
	[41]	ISH	✗	✗	✓	In Drosophila gene patterns, texture features are effective	81.90%
2010	[42]	FISH	✓	✓	✓	Discriminative features, i.e., nucleus shape and texture, are used for the final detection of leukemia	95.00%
2011	[43]	FISH	✓	✓	✓	The contour signature and Hausdorff Dimensions are used for classifying a lymphocytic cell	93.00%
2012	[44]	FISH	✓	✗	✗	Spindle-shaped features are extracted for the classification of FISH cells	–
	[45]	M-FISH	✓	✓	✗	Multicolor sparse imaging representation approach based on L1-norm minimization	90.00%
	[10]	ISH	✗	✗	✓	Local binary patterns or histograms are used to train the gene classifiers based on four cerebellum layers	94.00%
2014	[46]	Stained Blood Images	✓	✓	✗	A quantitative microscopic method is used for determination of lymphoblasts	90.00%
	[47]	Hyper spectral images	✗	✗	✓	GLCM texture features are used for hyper spectral images (HSIs)	–
2015	[48]	ISH	✓	✓	✗	Nuclei are segmented using k-means. Then, statistical and geometric features are used for cell classification using an SVM	98%
	[49]	Hyper spectral images	✗	✗	✓	Eight texture statistical features based on gray-level co-occurrence matrix (GLCM)	71.8%
2016	[50]	Tissue images	✗	✓	✗	Patch samples are selected based on stains on density maps with stain color	–
	[41]	ISH	✗	✗	✓	Image pixel-based DCNN is used for feature extraction	81.00%
2018	[51]	DICOM files	✓	✓	✓	The shape, gray-level co-occurence matrix, gray-level run length matrix, and neighborhood intensity difference were used to extract 386 texture features	80.39%
2019	[52]	FISH	✗	✗	✓	In total, 279 textural features and a machine learning classifier-based method were used	86.00%
2020	[53]	Blood Smear Images	✓	✓	✗	Different shades of color and brightness levels are computed from blood smears, and then the classifiers were applied	98.80%
	[54]	FISH	✗	✗	✓	In total, 488 texture features were extracted from precontrast, postcontrast, and subtraction images	83.00%
2021	[55]	Microscopy	✗	✓	✓	Homogeneous regions were segmented using clustering techniques in the RGB color space	90%

Note: SF stands for shape features, CF stands for color features, and TF stands for texture features. The check and cross symbols indicate that the features belong to the corresponding method and reference.

**Table 5 diagnostics-14-02089-t005:** Explanations of segmentation methods used in digital pathology for nuclei and cell segmentation.

Year	Pathology Image Type	Application	Segmentation Technique	Ref.
**Nuclei Segmentation ↓**				
2009	Cervical tissue	Region-based segmenation	Clustering method is used in RGB color space for nuclei segmentation	[55]
2010	FISH	Nuclei segmentation	Morphologial image enhancement and watershed technique	[62]
2012	SISH	*HER2* gene status	Number of cells, genes, number of genes per cell (average), superimposed contour cell image, gene image, and processing time	[55]
2016	FISH	*HER2* gene status	A method for nuclei segmentation from the blue channel of the contrast-enhanced image	[63]
2018	FISH	Segmentation and detection of signals	Enhanced nucleus segmentation and signal detection from tile-based processing using the adaptive thresholding	[64]
2019	FISH	Segmenation and classification	Two RetinaNet networks for the detection and classification of nuclei into distinct classes and classifing FISH signals into HER2 or CEN17	[65]
2020	IHC	Machine learning-based segmentation	Annotated dataset for training machine learning techniques, which includes firmly packed nuclei from several tissues	[66]
**Cancer cell detection ↓**				
2015	Microscopy Images	Fast characterization of apoptotic cells	Adaptive thresholding, a support vector machine, a majority vote, and the watershed technique are used	[67]
**Tumor area detection ↓**				
2021	FISH	Three-dimensional scoring of fluorescence	Three-dimensional FISH scoring is established for automated z-stack images from confocal WSI scanner	[68]

## Data Availability

No new data were created or analyzed in this study. Data sharing is not applicable to this article as it is a review based on previously published studies and publicly available datasets.

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
