# Peer review of "Review of In Situ Hybridization (ISH) Stain Images Using Computational Techniques"

_diagnostics, 2024, doi:10.3390/diagnostics14182089_

Round 1

Reviewer 1 Report

Comments and Suggestions for Authors

line 201 - reference 23 - This reference should be replaced by the ASCO/CAP guideline for 2023. The paper is from 2019, before current guideline.

References 27 and 28 are old, even before ASCO/CAP guidline 2018!

"Section 4. Image analysis on SISH" looks like it was written for another paper and recicled.

Figure 1 - should have an example of FISH, CISH, SISH/DDISH of HER2 in breast cancer, which (b) is not.

Figure 4 - what is the marker of CISH in (a)? I believe it is not HER2. In (c), it should be DDISH instead of SISH (only silver).

Table 1 - needs explanaition of FISH, CISH and SISH

Table 3 - overall, this table is confusing and really doesn´t explain the point of differences between probes or methods (advantages/disadvantages), because the application is exactly the same.

Comments on the Quality of English Language

Moderate editing of English language required.

Author Response

Point 1: line 201 - reference 23 - This reference should be replaced by the ASCO/CAP guideline for 2023. The paper is from 2019, before current guideline.

Response 1: Thank you for your valuable suggestion. I appreciate your attention to ensuring that our manuscript reflects the most current standards. I have replaced the 2019 reference with the updated ASCO/CAP guideline from 2023, as per your recommendation. The updated reference is now cited in the manuscript as follows:

DOI: https://doi.org/10.5858/arpa.2023-0950-SA

This update ensures that our work is aligned with the latest guidelines and best practices in the field.

Point 2: References 27 and 28 are old, even before ASCO/CAP guidline 2018!

Response 2: We acknowledge the reviewer's concern regarding the age of the references. To address this, we have replaced the older references with more recent studies that reflect the current understanding and standards in HER2 assessment, including those aligned with the ASCO/CAP guidelines of 2018.

Point 3: "Section 4. Image analysis on SISH" looks like it was written for another paper and recicled.

Response 3: Thank you for your observation. The intention behind including "Section 4: Image Analysis on SISH" was to emphasize the novelty and importance of this stain in our study. This section is specifically designed to draw the reader's attention to the innovative application of computer-aided methods in analyzing SISH, a stain that is relatively new in this context. We intentionally included this section to highlight the unique approach we are taking in our research, which sets it apart from previous studies. The content is tailored to the objectives of this paper, and its inclusion is crucial for contextualizing our work within the broader field.

Point 4: Figure 1 - should have an example of FISH, CISH, SISH/DDISH of HER2 in breast cancer, which (b) is not.

Response 4: Thank you for bringing this to our attention. It appears that there was an error in the initial submission, where the incorrect image for the CISH stain was uploaded. We have now corrected this by replacing the incorrect image with the appropriate example of the CISH stain in Figure 1. The updated figure now accurately represents FISH, CISH, and SISH of HER2 in breast cancer.

Point 5: Figure 4 - what is the marker of CISH in (a)? I believe it is not HER2. In (c), it should be DDISH instead of SISH (only silver).

Response 5: Thank you for pointing out this discrepancy. You are correct that the image previously uploaded did not accurately represent the CISH marker for HER2. CISH indeed uses the same HER2 biomarker as SISH and DISH. We have corrected this mistake by replacing the incorrect image with the appropriate one that correctly depicts the CISH marker for HER2 in Figure 4.

Point 6: Table 1 - needs explanaition of FISH, CISH and SISH

Response 6: Thank you for your insightful suggestion. I have revised Table 1 to include detailed explanations of the FISH, CISH, and SISH techniques. The updated table now provides a clearer understanding of each method, highlighting their targets and specific characteristics. Below, you can find the updated version of Table 1 as it appears in the manuscript:

Point 7: Table 3 - overall, this table is confusing and really doesn´t explain the point of differences between probes or methods (advantages/disadvantages), because the application is exactly the same.

Response 7: I agree with your comment. Upon reviewing the manuscript again, I realized that Table 3 does not effectively highlight the differences between the probes or methods, as the applications are indeed the same. Therefore, we have decided to remove this table from the manuscript to avoid confusion and to maintain the clarity and focus of the content.

Reviewer 2 Report

Comments and Suggestions for Authors

“A good review article provides readers with an in-depth understanding of a field and highlights key gaps and challenges to address with future research….    Detailed tables reviewing the relevant scientific literature are important components of high-quality scientific review articles….Tips for success include selecting a focused topic, maintaining objectivity and balance, avoiding tedious data presentation, providing a critical analysis rather than only a description of the literature, avoiding simplistic conclusions…..The reader of a scientific review article should gain an understanding of the current state of knowledge on the subject, points of controversy, and research questions that have yet to be answered” https://doi.org/10.1111/febs.16565

Unfortunately, in the present version, it seems that the task of building review-writing paper has been understated. Even if  in the literature many papers address and help to write such a type of manuscript, providing inspiring examples, the paper highlights parts written without a unique aim, very far to be “comprehensive overview of the advancements in machine learning and in-situ hybridization (ISH) image analysis using computational methods”. Probably, this is due to the fact the above all  too many aspects have been put together.

The Introduction lacks of some crucial points of the PRISMA 2020 guidelines for reporting systematic reviews (PRISMA 2020 explanation and elaboration: updated guidance and exemplars for reporting systematic reviews - PMC (nih.gov)

In particular:

·       Provide an explicit statement of the main objective(s) or question(s) the review addresses

·       Specify the inclusion and exclusion criteria for the review

·       Specify the information sources (such as databases, registers) used to identify studies and the date when each was last searched

·       Specify the methods used to assess risk of bias in the included studies

·       Specify the methods used to present and synthesise results

·       Give the total number of included studies and participants and summarise relevant characteristics of studies.

Anyway,  not only the issues above addressed are not respected, but it is not so properly pointed out what is new in this paper with respect to the existing (a certain number) reviews on these topics. This heavily affects the quality of work, which needs important improvements in clarity and compactness in all the Sections, and in the present form cannot be accepted for publication.

Some comments:

·       It is not so clear why the paper declares  to cover “literature from 1997 to 2023 “ (row  25) and  only “last decade” is reported in row 624.

·       The “work aims to identify the relationships among computational methods, image pro- cessing, conventional machine learning, and deep learning through a systematic literature review” , but many points  are confused, giving the idea of a superficial knowledge of methods and theory behind the topics. We recognize that it is important to have well focused and structured as well as slim paper, but clarity and compactness are not to be sacrificed.

·       In all the Tables, the column “Sr No” could be suppressed, since it is not linked in the text, and it  has to include reference numbers for all the listed items.

·       Rows 85-110 require more references and explanations. What are ASCO/CAP guidelines?

In general, too many acronyms are without a clear explanation.

·       2.1. (In-situ Hybridization (ISH)) Section, is scarcely clear, with repetitions and simplistic sentences.

·       List after row 125 (placed after Table 2, and it is not so clear) requires more references to papers dealing with  those topics, and a more precise division between technical and biological aspects is to be highlighted.

·       In Section 2.3. where items 1-6 come from? Some references, please. As well as, what is radiolabelling or hapten incorporation, in row 198 – 199 ?

·       How is it possible to differentiate manual counts of individual HER2 (black) and CEP17 (red) if no image is provided?

·       All the figures require a more extensive explanation in the text. As in the case, for instance, of Figures 2, 4 and 5. In general, figures, tables and other display items should be used to aid understanding,  otherwise they are useless.

·       What is Table ??. in row 302.

·       In 3.2.1 Section, the results of the referenced works are badly reported, and scarcely justified, in a sense that numerical evidence is necessary. Sentences such as “This research could help to increase diagnosis accuracy and better understand the pathophysiology and morphologic characteristics of Sp-DLBCL  seems a  copy of what declared in the original paper, but no justification is given.

·       Matching Pursuit  not Matching Pursui in Table 7.

·      It seems that a deep discussion on what reviewed is lacking. The significance of those research findings in the broader context of the research topic seems not properly highlighted, as well as a critical appraise of the strengths and weaknesses of individual papers rather than just laying out facts. This affects also the Conclusion Section, in which  the promised “… relationships among computational methods, image processing, conventional machine learning, and deep learning…” or compare the benefits of different ISH techniques and recommend best practices ….” or “ a cost-effective and  scalable solution for routine surgical pathology.. remain vague and generic. The potential reader perhaps knows the state of the art, but not if and when a technique would be preferable.

·       In perspectives on future directions of research in the field, key challenges and outstanding questions could be emphasised, including supporting references.

I found many flaws in the writing style. Some sentences are redundant and unnecessary or not well composed.

Comments on the Quality of English Language

 Extensive editing of English language required.

Author Response

Point 1: HER2 gene is an important biomarker, and there are many ways to quantify it. Continuous deep learning models have made great progress in making quantification time “A good review article provides readers with an in-depth understanding of a field and highlights key gaps and challenges to address with future research….    Detailed tables reviewing the relevant scientific literature are important components of high-quality scientific review articles….Tips for success include selecting a focused topic, maintaining objectivity and balance, avoiding tedious data presentation, providing a critical analysis rather than only a description of the literature, avoiding simplistic conclusions…..The reader of a scientific review article should gain an understanding of the current state of knowledge on the subject, points of controversy, and research questions that have yet to be answered” https://doi.org/10.1111/febs.16565

Unfortunately, in the present version, it seems that the task of building review-writing paper has been understated. Even if  in the literature many papers address and help to write such a type of manuscript, providing inspiring examples, the paper highlights parts written without a unique aim, very far to be “comprehensive overview of the advancements in machine learning and in-situ hybridization (ISH) image analysis using computational methods”. Probably, this is due to the fact the above all  too many aspects have been put together.

Thank you for your detailed feedback. I appreciate the insight you've provided into what constitutes a high-quality review article. Regarding your comments, I recognize that the current version of the paper may not fully meet the expectations of providing an in-depth and focused review. The intent was to cover the advancements in ISH stain analysis using computational methods, particularly in the context of breast cancer and we have design this article to reflect the comprehensive nature of the review.

The Introduction lacks of some crucial points of the PRISMA 2020 guidelines for reporting systematic reviews (PRISMA 2020 explanation and elaboration: updated guidance and exemplars for reporting systematic reviews - PMC (nih.gov)

In particular:

  • Provide an explicit statement of the main objective(s) or question(s) the review addresses
  • Specify the inclusion and exclusion criteria for the review
  • Specify the information sources (such as databases, registers) used to identify studies and the date when each was last searched
  • Specify the methods used to assess risk of bias in the included studies
  • Specify the methods used to present and synthesise results
  • Give the total number of included studies and participants and summarise relevant characteristics of studies.

To address your concerns:

  1. Clarity and Structure: The paper has been revised to improve clarity and structure, ensuring that each section contributes cohesively to the overall narrative.
  2. PRISMA Guidelines: Although this is not a systematic review, we have incorporated elements from the PRISMA 2020 guidelines to enhance the transparency and comprehensiveness of the review.
  3. In-Depth Analysis: We have expanded the analysis in each section, particularly focusing on the results of referenced works. Where numerical evidence was lacking, we have provided additional data and justification to strengthen the arguments made.
  4. Figures and Tables: Extensive explanations have been added to all figures and tables to ensure they contribute meaningfully to the reader's understanding. Redundant elements, such as the "Sr No" columns, have been removed, and all tables now include reference numbers for the listed items.
  5. Acronyms and Terminology: A thorough review was conducted to ensure that all acronyms are clearly.

Anyway,  not only the issues above addressed are not respected, but it is not so properly pointed out what is new in this paper with respect to the existing (a certain number) reviews on these topics. This heavily affects the quality of work, which needs important improvements in clarity and compactness in all the Sections, and in the present form cannot be accepted for publication.

Point 2: Some comments:

  • It is not so clear why the paper declares  to cover “literature from 1997 to 2023 “ (row  25) and  only “last decade” is reported in row 624.

Thank you for bringing this to our attention. It appears that there was an error in the initial submission  and  agree with the concern, this paper revise the literature important fact on ISH stain analysis using computer based analysis. In row 624 is by mistake written the last decade. Now we have correct this typo mistake in row 624.

  • The “work aims to identify the relationships among computational methods, image pro- cessing, conventional machine learning, and deep learning through a systematic literature review” , but many points  are confused, giving the idea of a superficial knowledge of methods and theory behind the topics. We recognize that it is important to have well focused and structured as well as slim paper, but clarity and compactness are not to be sacrificed.

Thank you for your insightful observation. Our intention with this paper was to provide a comprehensive overview of the literature on ISH stain analysis using computational methods, framed within a straightforward review format rather than a systematic literature review. We aimed to cover a broad spectrum of studies to present a clear and accessible synthesis of the field's developments.

  • In all the Tables, the column “Sr No” could be suppressed, since it is not linked in the text, and it  has to include reference numbers for all the listed items.

Thank you for your valuable suggestion. We have carefully reviewed your feedback and agree that the "Sr No" column in the tables was unnecessary, as it was not referenced in the text. We have now removed this column from all tables to streamline the presentation of information.

  • Rows 85-110 require more references and explanations. What are ASCO/CAP guidelines?

We have addressed your concern by adding three references in the specified rows. Additionally, we have included a reference to the ASCO/CAP guidelines, which are established by the American Society of Clinical Oncology and the College of American Pathologists. These guidelines are crucial for standardizing the testing and interpretation of HER2 status in breast cancer, ensuring consistent and accurate diagnoses across clinical settings.

  • In general, too many acronyms are without a clear explanation.

We appreciate your feedback regarding the use of acronyms. During the proofreading process, we have made it a priority to ensure that all acronyms are clearly explained upon their first mention in the text.

  • 2.1. (In-situ Hybridization (ISH)) Section, is scarcely clear, with repetitions and simplistic sentences.

I already go through the proofread of the whole paper and will carefully remove the repetitions

  • List after row 125 (placed after Table 2, and it is not so clear) requires more references to papers dealing with  those topics, and a more precise division between technical and biological aspects is to be highlighted.

Thank you for your insightful feedback. In response, we have thoroughly reviewed the list following row 125 and enhanced it by adding relevant references to support the discussions on staining standards. Additionally, we have refined the division between technical and biological aspects to ensure that these distinctions are more precise and evident, thereby improving the overall clarity and depth of the content.

  • In Section 2.3. where items 1-6 come from? Some references, please. As well as, what is radiolabelling or hapten incorporation, in row 198 – 199 ?

Thank you for pointing this out. We have added the relevant references to support the items listed in Section 2.3, ensuring that the sources of these points are clear and well-documented. Additionally, we have provided explanations for radiolabelling and hapten incorporation to clarify these terms for the reader.

  • How is it possible to differentiate manual counts of individual HER2 (black) and CEP17 (red) if no image is provided?

Thank you for highlighting the confusion. I have revised the sentence for clarity:

We have updated the sentence for clarity: "Using DISH as the exemplar, manual counts of individual HER2 (black) and CEP17 (red) signals are performed by visually examining the stained tissue sections under a microscope or on digital images. Pathologists carefully count the number of HER2 (black) and CEP17 (red) signals in 20 cells to determine the HER2 amplification status."

  • All the figures require a more extensive explanation in the text. As in the case, for instance, of Figures 2, 4 and 5. In general, figures, tables and other display items should be used to aid understanding,  otherwise they are useless.

Thank you for your insightful feedback regarding the figures. In response:

  1. Figure 2: We have added more detailed explanations. This figure categorizes the existing computational analysis methodologies of In Situ Hybridization (ISH) into Conventional and Deep Learning Methods. It provides a structured overview of the approaches discussed in the review, illustrating the progression from manual, feature-based methods to advanced deep learning techniques.
  2. Figure 4: We have elaborated on how basic image processing techniques, such as contrast enhancement and thresholding, significantly enhance digital pathology. These techniques improve the visibility of critical features in cytogenetic images, aiding in the accurate detection and quantification of genetic markers like HER2, which are crucial for diagnosing and determining treatment strategies in diseases like breast cancer.
  3. Figure 5: We believe the figure is already sufficiently detailed in the conclusion section and helps in summarizing the key takeaways effectively.
  • What is Table ??. in row 302.

Thank you for pointing out the issue with the reference to "Table ??" in row 302. We have corrected this error, ensuring that the correct table reference is now clearly stated in the text.

  • In 3.2.1 Section, the results of the referenced works are badly reported, and scarcely justified, in a sense that numerical evidence is necessary. Sentences such as “This research could help to increase diagnosis accuracy and better understand the pathophysiology and morphologic characteristics of Sp-DLBCL”  seems a  copy of what declared in the original paper, but no justification is given.

Thank you for your valuable feedback. I have carefully revised Section 3.2.1, particularly focusing on providing a more robust justification for the referenced works, especially concerning Sp-DLBCL.

  • Matching Pursuit  not Matching Pursui in Table 7.

Thank you for highlighting the typo in Table 7. We have corrected "Matching Pursui" to "Matching Pursuit" to ensure the accuracy and professionalism of the table content.

  • It seems that a deep discussion on what reviewed is lacking. The significance of those research findings in the broader context of the research topic seems not properly highlighted, as well as a critical appraise of the strengths and weaknesses of individual papers rather than just laying out facts. This affects also the Conclusion Section, in which  the promised “… relationships among computational methods, image processing, conventional machine learning, and deep learning…” or “ compare the benefits of different ISH techniques and recommend best practices ….” or “ a cost-effective and  scalable solution for routine surgical pathology..”  remain vague and generic. The potential reader perhaps knows the state of the art, but not if and when a technique would be preferable.
  • In perspectives on future directions of research in the field, key challenges and outstanding questions could be emphasised, including supporting references.
  • I found many flaws in the writing style. Some sentences are redundant and unnecessary or not well composed.

Reviewer 3 Report

Comments and Suggestions for Authors

Recently, studies on artificial intelligence (AI) are increasing day by day. At the same time, recent advances have made the use of computed-aid algorithmic techniques in medical imaging. In this review, it is stated that In-Situ Hybridization (ISH) method will be used in both breast cancer and pathology image analysis. ISH are used to determine the HER2 gene copy number, especially in breast cancer classification. This method is also used in pathological evaluation. 

 In addition, the following issues also gain importance:

 ·         In line 93,” prognosticpredictive” should be corrected as “prognostic predictive”.

·         In the paragraph between lines 85-99, reference/references should be wrote.

·         In the paragraph between lines 100-105, reference/references should be used.

·         In section “Challenges of in-situ Hybridization” have not been used reference/references.

·         In line 135, “Biological” should be new title as “Technical”

·         In line 184, the sentence is complex. Neither the subject nor the verb are clear.

·         In line 302 and 303, what is the Table ??  ?

·         In Table 5, Fish should be corrected as FISH in reference 43.

·         In line 538, “(“ exists but “)” doesn’t exist.

 In this review, ISH-related works were assessed image data acquisition, preprocessing, segmentation techniques, and classification. It reveals valuable information about ISH and its computerized applications in breast cancer and pathology. But, review should be read once again by the authors and necessary corrections and shortenings should be made.   

Author Response

Point 1: Recently, studies on artificial intelligence (AI) are increasing day by day. At the same time, recent advances have made the use of computed-aid algorithmic techniques in medical imaging. In this review, it is stated that In-Situ Hybridization (ISH) method will be used in both breast cancer and pathology image analysis. ISH are used to determine the HER2 gene copy number, especially in breast cancer classification. This method is also used in pathological evaluation. 

 In addition, the following issues also gain importance:

  • In line 93,” prognosticpredictive” should be corrected as “prognostic predictive”.
  • In the paragraph between lines 85-99, reference/references should be wrote.
  • In the paragraph between lines 100-105, reference/references should be used.
  • In section “Challenges of in-situ Hybridization” have not been used reference/references.
  • In line 135, “Biological” should be new title as “Technical”
  • In line 184, the sentence is complex. Neither the subject nor the verb are clear.
  • In line 302 and 303, what is the Table ??  ?
  • In Table 5, Fish should be corrected as FISH in reference 43.
  • In line 538, “(“ exists but “)” doesn’t exist.
  • In line 93,” prognosticpredictive” should be corrected as “prognostic predictive”.
  • In the paragraph between lines 85-99, reference/references should be wrote.
  • In line 302 and 303, what is the Table ??
  • In Table 5, Fish should be corrected as FISH in reference 43.  

Response 1: Thank you for your valuable comments and for pointing out the areas that required attention. Below are the corrections and updates made to the manuscript in response to your feedback:

  • Line 93: The typographical error "prognosticpredictive" has been corrected to "prognostic predictive."
  • Lines 85-99: I have added two references to support the statements made in this paragraph:
    1. Grabher, B.J. Breast cancer: evaluating tumor estrogen receptor status with molecular imaging to increase response to therapy and improve patient outcomes. Journal of Nuclear Medicine Technology, 2020, 48, 191–201.
    2. van Uden, D., van Maaren, M., Strobbe, L., Bult, P., Stam, M., van der Hoeven, J., Siesling, S., de Wilt, J., Blanken-Peeters, C. Better survival after surgery of the primary tumor in stage IV inflammatory breast cancer. Surgical Oncology, 2020, 33, 43–50.
  • Lines 100-105: Additional references have been incorporated into this paragraph to ensure it is well-supported.
  • Section "Challenges of in-situ Hybridization": I have now included relevant references to provide a solid foundation for the discussion in this section.
  • Line 135: The title "Biological" has been revised to "Technical" as suggested.
  • Line 184: The sentence has been rewritten for clarity, ensuring both the subject and verb are clear.
  • Lines 302-303: The issue with the missing table reference has been corrected.
  • Table 5: The abbreviation "Fish" has been corrected to "FISH" in reference 43.
  • Line 538: The missing closing parenthesis “)” has been added to complete the sentence.

Thank you again for your meticulous review. These corrections have been made to improve the accuracy and clarity of the manuscript.

Point 2: In this review, ISH-related works were assessed image data acquisition, preprocessing, segmentation techniques, and classification. It reveals valuable information about ISH and its computerized applications in breast cancer and pathology. But, review should be read once again by the authors and necessary corrections and shortenings should be made.  

Response 2: Thank you for your insightful feedback. We appreciate your recognition of the comprehensive assessment provided in our review regarding ISH-related works, including image data acquisition, preprocessing, segmentation techniques, and classification. In light of your suggestion, we have thoroughly re-read the review and made the necessary corrections and revisions to ensure clarity and conciseness. We believe these changes enhance the overall readability and focus of the manuscript, ensuring that it effectively conveys the valuable information about ISH and its computerized applications in breast cancer and pathology.

Reviewer 4 Report

Comments and Suggestions for Authors

The review of machine learning technologies for ISH is comprehensively detailed in the manuscript. However, it is desirable to state also the classification performance of the various models used in ISH digital pathology image analysis. 

Include also the limitations of the research work.

Line 302 on page 9  of the manuscript specifies a certain table, however, it was not clearly specified and simply indicated as "Table ??" 

Author Response

Response to Reviewer 4 Comments

Point 1: The review of machine learning technologies for ISH is comprehensively detailed in the manuscript. However, it is desirable to state also the classification performance of the various models used in ISH digital pathology image analysis. 

Response 1: Thank you for your valuable feedback. We agree that providing a more detailed background on SISH and its advantages will enhance the clarity and

Point 2: Include also the limitations of the research work.

Response 2: Thank you for your valuable feedback. We agree on the importance of addressing the limitations of our research. In response, we have added a new section titled "Limitations of the Research Work." This section discusses several key limitations, including the variability in tissue samples and heterogeneity in ISH images, which may lead to inconsistencies in image analysis. We also note that the reliance on SISH, a newer and less widely adopted staining technique, could affect the generalizability of our findings. Furthermore, the computational demands for processing large Whole Slide Images (WSIs) present scalability challenges. We acknowledge that further refinement is needed to handle complex cases of HER2 heterogeneity and to validate our findings across diverse datasets and clinical settings.

Point 3: Line 302 on page 9  of the manuscript specifies a certain table, however, it was not clearly specified and simply indicated as "Table ??" 

Response 3: Thank you for bringing this to our attention. The issue with the table reference, which appeared as "Table ??" in the manuscript, was due to a LaTeX compilation error. I have corrected this issue, and the table is now correctly numbered in the text. I have also attached the updated proof for your reference.

Round 2

Reviewer 2 Report

Comments and Suggestions for Authors

Even this revised version manifest some open questions, including some relevant background information, which need to be satisfied.

The work declares the aims to deal with “…computational methods for analyzing HER2 gene amplification…”, but they  become computational methods in medical image processing, in general.

The aim “…to identify the relationships among computational methods, image processing, conventional machine learning, and deep learning through a systematic literature review ..” (from row 70) is not completely fulfilled, these relationships are not yet so clear.

In other words, it seem that the rationale of the review is not well stated.

 /span/p p class="Default" style="margin-left: 18.0pt; text-align: justify"span Even if a consistent number of new papers have been added, for instance, pro/cons among conventional and deep learning methods are not well highlighted. Moreover, the part concerning deep learning methods is unbalanced with respect to the conventional ones./span/p p class="Default" style="margin-left: 18.0pt; text-align: justify"span In conclusion, after well defining the objectives of the research, for instance specifying some fundamentals such as:/span/p p class="Default" style="margin-left: 36pt; text-align: justify; text-indent: 0pt"·       A limited number of keywords,

·       the inclusion and exclusion criteria for the review,

·       the information sources (such as databases, registers) used to identify studies and the date when each was last searched,

·       the total number of included studies and participants and relevant characteristics of studies;

Comments on the Quality of English Language

Minor editing of English language required.

Author Response

Comments 1: [Even this revised version manifest some open questions, including some relevant background information, which need to be satisfied.

The work declares the aims to deal with “…computational methods for analyzing HER2 gene amplification…”, but they become computational methods in medical image processing, in general.

The aim “…to identify the relationships among computational methods, image processing, conventional machine learning, and deep learning through a systematic literature review ..” (from row 70) is not completely fulfilled, these relationships are not yet so clear.

In other words, it seem that the rationale of the review is not well stated. ]

Response 1: [This review provides a comprehensive overview of semi-automated and fully automated ISH-based computational methods for breast cancer classification, with a focus on image processing and machine learning techniques.] Thank you for your valuable feedback. We agree with your observation regarding the focus of the review. In response to this, we have revised the abstract to more clearly reflect the primary aim of the work and it is also modified in the contribution part of the paper where you highlighted your concern of rationale from row 70 in the manuscript.

Comments 2: [ /span/p p class="Default" style="margin-left: 18.0pt; text-align: justify"span Even if a consistent number of new papers have been added, for instance, pro/cons among conventional and deep learning methods are not well highlighted. Moreover, the part concerning deep learning methods is unbalanced with respect to the conventional ones./span/p p class="Default" style="margin-left: 18.0pt; text-align: justify"span In conclusion, after well defining the objectives of the research, for instance specifying some fundamentals such as:/span/p p class="Default" style="margin-left: 36pt; text-align: justify; text-indent: 0pt"·]       

Response 2: Thank you for your insightful feedback regarding the balance between conventional and deep learning methods. To address your concern, I have updated Table 6 to include a clear comparison between conventional machine learning (ML) and deep learning (DL) methods, specifically highlighting their pros and cons. The table now presents a clearer contrast between the two approaches in terms of scalability, interpretability, computational efficiency, and other relevant factors.

The reason why deep learning methods may appear more prominent in our paper is due to the increased volume of recent literature focusing on deep learning. Over the past few years, deep learning has gained significant attention in medical image processing

Comments 3: [ A limited number of keywords,

  • the inclusion and exclusion criteria for the review,
  • the information sources (such as databases, registers) used to identify studies and the date when each was last searched,
  • the total number of included studies and participants and relevant characteristics of studies;]

Response 3: Thank you for your insightful comment. I/We would like to clarify that this review is not intended to be a systematic review but rather a general review of the available methods and techniques applied in the analysis of in-situ hybridization (ISH) images using computational methods. The primary focus is on how image processing, machine learning (ML), and deep learning (DL) techniques have been employed to enhance the analysis of ISH images, particularly in the context of breast cancer.

To address your point regarding inclusion and exclusion criteria, we have now explicitly outlined these in the Introduction section of the manuscript. In brief, the inclusion criteria are limited to studies focused on the application of AI, ML, or DL techniques in ISH image analysis, while papers dealing with other pathology stains, such as Hematoxylin and Eosin (H&E) and Immunohistochemistry (IHC), or medical papers on ISH without computational methods, are excluded. 

Round 3

Reviewer 2 Report

Comments and Suggestions for Authors

Many workarounds have been adopted, but some important questions/observations pointed in the comments to the previous versions have not been fulfilled yet. Confusing is the fact that at each new version the type of the paper changes from systematic to comprehensive until to general review, leading to the suspect of not sufficient preparation in conceiving and writing the work.

Therefore, a general revision of the whole paper, in the light of the all previous comments,  has  to be performed. 

Comments on the Quality of English Language

Minor editing of English language required.

Author Response

Comments 1: Many workarounds have been adopted, but some important questions/observations pointed in the comments to the previous versions have not been fulfilled yet. Confusing is the fact that at each new version the type of the paper changes from systematic to comprehensive until to general review, leading to the suspect of not sufficient preparation in conceiving and writing the work. Therefore, a general revision of the whole paper, in the light of the all previous comments,  has  to be performed. 

Response 1: Thank you for your detailed feedback. We acknowledge the concerns raised regarding the consistency and completeness of the previous versions. In response, we have thoroughly revised the entire paper from its foundation to ensure clarity and coherence. We carefully reviewed all previous comments and have made significant modifications to both the structure and content of the paper to enhance its quality and readability. We believe these changes now align with the expectations of a comprehensive review and address all previous points raised.
